# Thermal and Chemical Characterization of Kenaf Fiber (*Hibiscus cannabinus*) Reinforced Epoxy Matrix Composites

**DOI:** 10.3390/polym13122016

**Published:** 2021-06-20

**Authors:** Thuane Teixeira da Silva, Pedro Henrique Poubel Mendonça da Silveira, Matheus Pereira Ribeiro, Maurício Ferrapontoff Lemos, Ana Paula da Silva, Sergio Neves Monteiro, Lucio Fabio Cassiano Nascimento

**Affiliations:** 1Department of Materials Science, Military Institute of Engineering—IME, Praça General Tibúrcio 80, Urca, Rio de Janeiro 22290-270, Brazil; thuaneteixeiraa@gmail.com (T.T.d.S.); pedroo.poubel@gmail.com (P.H.P.M.d.S.); mpmatheusr@gmail.com (M.P.R.); lucio_coppe@yahoo.com.br (L.F.C.N.); 2Brazilian Navy Research Institute—IPqM, Materials Technology Group; Rua Ipiru, 02, Cacuia, Rio de Janeiro 21931-095, Brazil; mauricio.lemos@marinha.mil.br (M.F.L.);

**Keywords:** natural fibers, kenaf, epoxy composite materials, thermal degradation properties, chemical characterization

## Abstract

Kenaf (*Hibiscus cannabinus L.*) is one of the most investigated and industrially applied natural fibers for polymer composite reinforcement. However, relatively limited information is available regarding its epoxy composites. In this work, both thermal and chemical properties were, for the first time, determined in kenaf fiber reinforced epoxy matrix composites. Through XRD analysis, a microfibrillar angle of 7.1° and crystallinity index of 44.3% was obtained. The FTIR analysis showed the functional groups normally found for natural lignocellulosic fibers. TMA analysis of the composites with 10 vol% and 20 vol% of kenaf fibers disclosed a higher coefficient of thermal expansion. The TG/DTG results of the epoxy composites revealed enhanced thermal stability when compared to plain epoxy. The DSC results corroborated the results obtained by TGA, which indicated a higher mass loss in the first stage for kenaf when compared to its composites. These results might contribute to kenaf fiber composite applications requiring superior performance.

## 1. Introduction

Sustainable composites reinforced with natural fibers are drawing increasing attention in both research works [1,2,3,4,5,6,7] and industrial applications [8,9,10,11,12,13,14]. This recognition is due to the natural fiber characteristics, such as cost effectiveness, biodegradability, recyclability, and renewability. Relevant properties are associated with natural fiber composites (NFCs), such as a high degree of flexibility, hygroscopicity, and reduced energy consumption for production due to less abrasiveness to processing equipment.

NFCs specific mechanical properties are comparable to those of synthetic fiber composites like fiberglass, as the density of natural lignocellulosic fibers (<1.6 g/cm^3^) is significantly lower than that of glass fiber (2.58 g/cm^3^) [15]. In particular, “Life Cycle Engineering” [16] is able to show the advantages of considering NFCs in terms of environmental and financial impact for designers and engineers to include these materials in future projects emphasizing sustainability strategies.

In this century, kenaf fiber has been one of the most investigated and industrially applied natural lignocellulosic fibers for reinforcement of polymer composites [17]. It has recently been investigated as a possible reinforcement of sustainable composites even as short discontinuous fibers [18]. The kenaf fiber is extracted from *Hibiscus cannabinus*, illustrated in Figure 1, an herbaceous plant from the family of *Malvaceae*, native to Africa and India. The kenaf crop is resistant to semiarid conditions such as those found in the Mediterranean region, which allows achieving significant above-ground production rates with a consumption of only 500–600 mm of rain water in 6 months [17]. 

Various commercial products such as high-quality papers, biocomposites for car door trimmings, interior shelving, bioplastics, building materials like medium-density fiberboard, textile, furniture, and many others have already been developed based on the physical and mechanical performance of kenaf fibers [19].

Although the kenaf fiber presents recognized disadvantages inherent to natural lignocellulosic fibers, it is comparatively a promising reinforcement for polymer composites. The cellulose content (31%– 72%) and microfibrillar angle (9°– 15°) influence the mechanical properties of the kenaf fiber, leading to relatively higher tensile strength [17,19]. As such, it was found that polypropylene (PP) composites reinforced with kenaf fiber display superior tensile and flexural strength when compared with those of other PP composites reinforced, for example, with hemp, sisal, and even glass fiber. This led to its application as an automotive structural component [20]. 

Actually, kenaf-reinforced composites have used a diversified number of polymer matrices including PP [21,22,23,24,25,26,27,28,29], polyester [30,31,32,33,34,35,36], polyurethane [37,38], high-density polyethylene [22,39,40,41], polystyrene [42], polylactic acid [43,44,45,46,47,48,49,50,51], natural rubber [52], PVC [53], and epoxy [54] among others. 

In particular, epoxy matrix reinforced with kenaf fiber was investigated for different purposes. Park et al. [54] evaluated the micromechanical properties of epoxy/kenaf composites and the wettability of kenaf fibers using non-destructive acoustic emission. Keshk et al. [55] performed physicochemical characterization of kenaf fibers to identify their chemical composition (cellulose, hemicellulose, and lignin) and their physical properties (viscosity, degree of polymerization, and crystallinity). 

Sgriccia and Hawley [56] reported the characterization of epoxy composites with 15 wt% of kenaf fiber to verify the effectiveness of curing the composite using the microwave method. Chin et al. [57] investigated the potential of vacuum-processed epoxy/kenaf composites in tribological application. Xue et al. [58] reported on improved mechanical properties (modulus of elasticity, tensile strength) of epoxy composites reinforced with kenaf Liberian fibers. Abu Bakar et al. [59] investigated the mechanical and morphological properties of epoxy composites reinforced with kenaf fiber treated through mercerization with NaOH. Mutasher et al. [60] showed the effect of alkaline mercerization treatment on kenaf fibers on the tensile strength and flexural strength of epoxy/kenaf composites. Abdullah et al. [61] reported on the fatigue behavior of unidirectional epoxy/kenaf composites prepared by hand lay-up. Suriani at al. [62] detected defects in kenaf/epoxy composite using infrared thermal imaging. 

It is also worth mentioning the investigation by Davoodi et al. [63] on the mechanical properties of hybrid kenaf/glass-reinforced epoxy composite for application in car bumper beams. Ali Kandemir et al. [18] investigated the adhesion, physical, and mechanical properties of epoxy matrix composites reinforced with discontinuous kenaf fibers, where their performance was compared with other fibers.

Despite these valuable research works, to our knowledge, no systematic investigation has been conducted, so far, on the combined thermal and chemical behavior of kenaf fiber reinforced epoxy composites to establish technical limits for possible industrial applications. The limited information on epoxy composites might be attributed to the relatively higher cost as compared to the other polymer resins. However, for high-performance applications, epoxy provides good adhesion to the reinforcing filler and improved enhanced mechanical properties in association with its low moisture content, less shrinkage upon curing, and easy room temperature processing [64].

Motivated by the aforementioned current status, the present work investigates the thermal and chemical characterization of epoxy composites reinforced with up to 30 vol% of continuous and aligned kenaf fibers.

## 2. Materials and Methods

### 2.1. Materials

The kenaf fibers illustrated in Figure 2 were supplied by Tapetes São Carlos Ltd., São Paulo, Brazil. 

Table 1 presents the basic properties of the as-supplied kenaf fibers applied in the present work to reinforce epoxy composites. The density of the kenaf fibers was obtained by calculating the mass using an analytical scale model AG-200 from Gehaka (São Paulo, Brazil) with a precision of 0.0001g divided by the volume (π D² / 4 × 150) of 100 fibers. The fiber diameter (D) values were measured by visualization under the profile projector of an optical microscope model BX53M from Olympus (Münster, Germany).

The results of the fiber density and diameter measurements are further shown in Section 3.1, and the mechanical properties were provided by the supplier. A commercial epoxy resin, diglycidyl ether of bisphenol A (DGEBA)-type hardened with triethylenetetramine (TETA) in a 13 phr stoichiometric ratio, fabricated by Dow Chemical, São Paulo, Brazil, and distributed by Resinpoxy Ltd, Rio de Janeiro, Brazil, was used as a polymeric matrix. Related information on DGEBA/TETA epoxy may be assessed at [65,66,67].

### 2.2. Composites Processing

The as-received kenaf fibers (Figure 2a) were first cleaned by immersing in water for 24 h to remove impurities. After immersion, the fibers were dried in a stove at 70 °C for 24 h. The fibers were separated and cut in a length of 150 mm (Figure 2b) for proper insertion in the mold for composite fabrication.

The composite plates production was done by compression, using a steel mold with rectangular dimensions of 150 × 120 × 12 mm^3^. The kenaf fibers were unidirectionally laid into the mold with the still liquid epoxy-hardener mixture in a previously calculated fiber-to-resin ratio. DGEBA/TETA epoxy composites incorporated with 10, 20, and 30 vol% of kenaf fiber, as shown in Figure 3, were fabricated using the epoxy and fiber densities listed in Table 1 to precisely determine the corresponding volume fractions. A plain epoxy plate (Figure 3a) was also fabricated as control. The composite plates were maintained under a 5-ton load in a SKAY hydraulic press (São Paulo, Brazil) for 24 h of curing. 

Although the composite plates in Figure 3b–d might not show perfectly aligned kenaf fibers, based on our experience, this is the best unidirectional alignment obtained by hand laying of natural fibers in a 150 mm long mold. During the introduction of fluid epoxy, it is typical for some fibers to slightly deviate from the alignment due to fluid flow.

### 2.3. Characterization

#### 2.3.1. Thermogravimetric Analysis (TG/DTG)

The thermogravimetric analyses of the epoxy, kenaf fibers, and the composites were performed in a Shimadzu DTG-60H machine (Tokyo, Japan). The samples were carefully hand-crushed with a pestle/mortar and then were placed in a platinum crucible. The procedure was performed under a nitrogen atmosphere with a heating rate of 10 °C/min, from 20 °C to 700 °C for the kenaf fiber and epoxy, as well as from 20 °C to 900 °C for the composites, according to ASTM E1131 [68]. As aforementioned in Section 2.2, prior to TGA analysis, the as-supplied kenaf fibers were cleaned and dried to remove impurities. However, owing to their hydrophilic behavior, the fibers surface immediately absorbs air humidity that appears in the TGA results.

#### 2.3.2. Fourier Transform Infrared Spectroscopy (FTIR)

FTIR analysis was performed to analyze the changes and interactions in the chemical bonding of kenaf fibers. FTIR spectra were recorded on NicoletiS10 equipment (ThermoFisher Scientific, Waltham, MA, USA) in a range of 500–4000 cm^−1^ with a resolution of 8 cm^−1^. The samples were comminuted and pressed under 10 kgf/cm pressure for 5 min, forming KBr pellets.

#### 2.3.3. Differential Scanning Calorimetry (DSC)

DSC tests were carried out in a DSC-60 machine from Shimadzu (Tokyo, Japan) under a nitrogen atmosphere with a heating rate of 10 °C/min and a temperature range from 20 °C to 400 °C for all the samples.

#### 2.3.4. X-ray Diffraction (XRD)

The XRD analysis was performed in a Shimadzu XRD-6000 diffractometer (Tokyo, Japan) with CuKα radiation, power of 1200 watts, −40 KV X 30 mA, and performing a 2θ scan from 5 to 80 degrees. The microfibril angle (MFA) was calculated using the methodology proposed by Donaldson [69] and Sarén and Serimaa [70] through the deconvolution of the crystalline cellulose peak (0 0 2). The calculation of fiber crystallinity follows the methodology presented by Segal et al. [71], where the maximum intensity obtained in the diffractogram is used for the area of the amorphous (1 0 1) and crystalline (0 0 2) peaks. These two peaks are associated with the amorphous (I_am_) and crystalline (I_crys_) phases, respectively. The crystallinity index of the fiber is calculated by:(1)$$\mathrm{CrI}=\left(\frac{I_{\mathrm{crys}}-I_{\mathrm{am}}}{I_{\mathrm{crys}}}\right)\;\times \;100\%$$

#### 2.3.5. Thermomechanical Analysis (TMA)

The measurements of the linear thermal expansion coefficient of the resin and the kenaf composite were performed with TMA-60 equipment (Shimadzu, Tokyo, Japan). The test was carried out according to ASTM E831 [72] under nitrogen atmosphere with a temperature range from 20 °C to 200 °C and a compression load of 10 gf.

## 3. Results and Discussion

### 3.1. Kenaf Fiber Density

The average density was obtained from the ratio between mass and volume in 100 as-supplied kenaf fibers. The mass of each fiber was determined on a precision scale. The corresponding fiber volume was calculated from its diameter measured in 10 points along its length in an optical microscope projector. For all 100 fibers, the average diameter was found as 71.2 ± 6.0 µm resulting in an average density of 1.52 ± 0.28 g/cm³.

### 3.2. Thermogravimetric Analysis (TGA)

Through TGA, it was possible to observe the mass loss process of the epoxy, kenaf fiber, and its composites with temperature. Figure 4 shows the TG/DTG curves for the DGEBA/TETA epoxy and the kenaf fiber. Beyond 700 °C, the TG curve remained horizontally constant, indicating no further mass loss.

In Figure 4a, a 1.65 % mass loss of the epoxy is observed in the first stage (up to 150°C), probably associated with the released humidity present in the synthetic polymer resin. The epoxy DTG curve displays prominent peaks associated with maximum mass loss rate at 307 °C and at 564 °C, which is attributed to subsequent depolymerization and degradation of the polymeric chains.

The TG curve of kenaf fiber, Figure 4b, is also composed of three stages, where the first (30 to 150 °C) is most probably related to the loss of surface humidity (about 9.80%). Natural fibers are known by its higher hydrophilicity. The greater degradation event of the fiber occurs in the second stage from 200 °C to about 570 °C, with a maximum mass loss rate at 355 °C associated with the loss of structural components of the fiber, mainly lignin, hemicellulose, and cellulose [73]. 

As for the DTG curve in Figure 4 (b), the small peak at 51 °C can be assigned to loss of kenaf fiber surface moisture, while the sharp peak at 355 °C and its shoulder around 300 °C are attributed to the degradation of cellulose and hemicellulose. The spread peak, as a bump around 500 °C–600 °C, might be related to lignin degradation [56].

Figure 5 shows the TG/DTG curves obtained for the kenaf composites. In the first degradation stage (25 to 150 °C), it is observed that the loss of mass increased with increasing fiber content, as expected, since epoxy presents a hydrophobic nature. 

Table 2 presents the main TGA parameters obtained for kenaf fiber, plain epoxy, and the investigated composites. The onset of thermal degradation (T_onset_) was obtained by the estimated point of deviation from the initial slope, almost a horizontal line in the TG curves of Figure 4 (epoxy and kenaf fiber) and Figure 5 (composites).

A minimum detected thermal degradation (T_min_) is proposed as the intercept between the initial, almost horizontal, slope and the steeper slope clearly related to the subsequent interval of thermal degradation of the material structure [73]. The point of maximun rate of thermal degradation (T_max_) is found as the DTG peak associated with the steeper region of mass loss, stage II on the TG curve.

Based on the values of T_onset_ in Table 2, it is possible to assume, in principle, that a safe maximum working temperature for the composite would be 200 °C, which is comparable to those of other natural fiber epoxy composites [73]. However, within the possible statistical variation of T_onset_ that is to be determined by a repeated number of TGA tests for several kenaf composite samples, the working temperature might eventually be safely raised to 220 °C as an upper limit.

In Table 2, it is worth noticing that enhancing the fiber content in the polymer is associated with a thermal stabilization of the composites. This is also corroborated by a higher residue content at 900 °C, especially for the composites with 20 vol% and 30 vol% kenaf fiber. The relatively small amount of char residue for both kenaf fiber and plain epoxy in Table 2 is an indication of near total carbonization at 700 °C. On the other hand, addition of kenaf fiber into the epoxy matrix enhances the composite thermal stability. This might be attributed to the fiber interaction, which is delaying the composite total degradation to higher temperatures. Indeed, at 900 °C, the composites char residues (3.19–9.10%) remain above those of kenaf fiber (2.14%) and epoxy (2.35%) found at 700 °C.

Regarding the results shown in Figure 6 and Table 2, one should bear in mind that incorporation of natural fibers into polymer matrices does not behave as an additive rule of mixture for thermal properties. The fiber interation with macromolecular chains affects the polymer glass transition of temperature, which is decreased, and then causes a total amorphous structure to be formed at lower temperature. This allows thermal softening to be displaced to higher temperatures. 

### 3.3. Fourier Transform Infrared Spectroscopy (FTIR)

Figure 6 depicts the vibrational spectrum obtained from the comminuted sample of the kenaf fiber.

Natural fibers have cellulose and lignin as main elements, which present a hydroxyl functional group in their structure. The FTIR spectrogram shows a band at 3352 cm^−1^ related to the stretching of the OH connection, and the 2917 cm^−1^ represents the elongation of the symmetrical and asymmetric CH bond. This is related to aliphatic groups rich in cellulose and hemicellulose. 

Another band at 1737 cm^−1^ corresponds to the ester carbonyl vibrations of the acetyl groups feruloyl and p-coumaril in lignin. An elongation band detected at 1650 cm^−1^ is assigned to the carbonyl group of the acetyl ester in hemicellulose and the carbonyl aldehyde in lignin. The broad band of 1244 cm^−1^ corresponds to the asymmetric C–O vibration [17,74,75]. 

The absorption band around 1053 cm^−1^ is related to the frequencies of C–O and C–C elongation of the xylans and elongation of the C–O–C glycosidic bond in hemicellulose. This band is also attributed to the C–O elongation, originating from the C-O-CH_3_ groups, which reinforces the existence of hemicellulose and lignin in the kenaf fibers [76].

### 3.4. Differential Scanning Calorimetry (DSC)

Figure 7 shows the DSC curves for the investigated materials. In Figure 7a, a DSC endothermic peak with a minimum temperature of 72 °C is observed in association with loss of moisture from the epoxy resin, which occurs at a temperature higher than that obtained by TGA. Moreover, two exothermic peaks at 324 °C and 352 °C can be attributed to the degradation and rupture of polymer chains, corroborating the TGA results.

The DSC curve for kenaf fiber (Figure 7b) presents an endothermic peak with a minimum at 80 °C, referring to the loss of moisture in the fiber. Even though the DSC moisture peak occurred at a significantly different temperature (51 °C) than that obtained by TGA in Figure 4b, both are within a similar temperature range. Indeed, TGA refers to weight loss, while DSC depicts heat flow. 

In both DTG (Figure 4b) and DSC (Figure 7b) curves, the kenaf fiber begins to lose mass and dissipate heat right from the beginning of the test at 20 °C. The maximum weight loss at 51 °C is mostly related to surface humidity release, and the DTG peak extends up to 100 °C, when total moisture is evaporated. On the other hand, the heat flow from water evaporation is cumulative and reaches a maximum at 80 °C, close to the point where all moisture is gone. The endothermic peaks in Figure 7(b) are correlated to the beginning of the decomposition of cellulose and hemicellulose (188 °C) and lignin (301 °C), which takes longer to start the decomposition.

The DSC curves obtained for the epoxy matrix composites with 10 vol%, 20 vol%, and 30 vol% kenaf are shown in Figure 7c. These curves corroborate what was observed in the TGA analysis, with the endothermic peak starting at 20 °C with minimum, at 63 °C for composites with 10 vol% and 20 vol% of fibers, and 70 °C for the 30 vol% composite. The peaks occurring at temperatures lower than those of the resin and fiber are due to the earlier loss of moisture in the composites.

### 3.5. X-ray Diffraction (XRD)

The XRD patterns were analyzed by computational treatment of the data using the OriginPro software. The diffractogram obtained for the kenaf fiber is shown in Figure 8.

For the determination of CrI, the peak intensity values associated with the amorphous and crystalline phases were used in Equation 1. Then, using the OriginPro software, normalization and derivation of the highest-intensity peak were performed for the calculation of the microfibrillar angle (MFA). 

The MFA of kenaf fibers was found to vary from 9 to 15° [19]. In this work, the MFA was 7.1°, a value that is lower than those found in the literature and close to ramie, hemp, and jute fibers. [77].

A comparatively lower MFA indicates a relatively stronger natural fiber [69]. The apparent difference between the MFA of 7.1° found in the present work, and that between 9° and 15° reported elsewhere [19], indicates that kenaf fiber from distinct origins (region, soil, crop, or season) might have different physical characteristics and mechanical properties. This is commonly verified in natural fibers of the same species [78].

The CrI for the present untreated kenaf fiber is only 44.3%, which implies a substantial presence of amorphous substances in the fiber.

### 3.6. Thermomechanical Analysis (TMA)

Figure 9 presents the TMA curves for the epoxy resin and the kenaf composites, as well as their glass transition temperatures (T_g_).

The glass transition temperatures obtained were 135 °C for the epoxy resin and 96, 94, and 120 °C for the composites with 10 vol%, 20 vol%, and 30 vol% kenaf reinforcement, respectively. Any factor that can affect the molecular mobility of the material, such as chemical composition, induction of crystallization by stretching, oxidation, addition of fillers, or a material that has a lower molecular mass, can affect the glass transition interval of the polymeric material and its thermal and mechanical behavior [79]. 

In the case of kenaf fibers, regarding the epoxy composite thermal behavior, the main factor is the addition of fibers as a reinforcing filler to the matrix, thus restricting the epoxy macromolecular mobility.

It was observed that the material undergoes expansion with increasing temperature. The linear thermal expansion coefficient obtained in this work was 112.38 × 10^−6^/°C for the epoxy resin and 181.16 × 10^−6^/°C, 187.02 × 10^−6^/°C, and 116.50 × 10^−6^/°C for the composites with 10 , 20 , and 30 vol% kenaf reinforcement, respectively.

## 4. Summary and Conclusions

Thermal and chemical properties were, for the first time, investigated in kenaf fiber reinforced epoxy matrix composites together with determination of fiber density and microfibrillar angle.

The DSC and TGA results indicated that there was an increase in the thermal stability of the composites with values of the onset (T_onset_), minimum detected thermal degradation (T_min_), and maximum rate of thermal degradation (T_max_) above those of the plain epoxy. Based on these values, a working temperature of 200 °C is proposed for the kenaf fiber epoxy composites. 

FTIR indicates the existence of functional OH, CH, C–O, and C–O–C bands that might be attributed to the kenaf fiber. These bands are known from fibers highly rich in lignin and hemicellulose. 

The XRD diffractogram showed characteristic peaks of lignocellulosic fibers. The presence of kenaf fiber peaks was detected at approximately 22° 2Ɵ, and through calculations, 44.3% fiber crystallinity index (CrI) and 7.1° microfibrillar angle (MFA) were identified. This MFA result might indicate a stronger mechanical property of the fiber. 

Through TMA analysis, it was observed that the addition of kenaf reinforcement at 10 vol% and 20 vol% decreased the T_g_ of the material, increasing its coefficient of thermal expansion. On the other hand, the 30 vol% composite presented a higher T_g_ and a lower coefficient of thermal expansion.

These results should contribute to the multiple applications expected for epoxy composites reinforced with kenaf fiber, which might display superior thermal stability, as well as comparatively stronger performance and lower thermal expansion, than other commonly used kenaf composites.

## Figures and Tables

**Figure 1 polymers-13-02016-f001:**
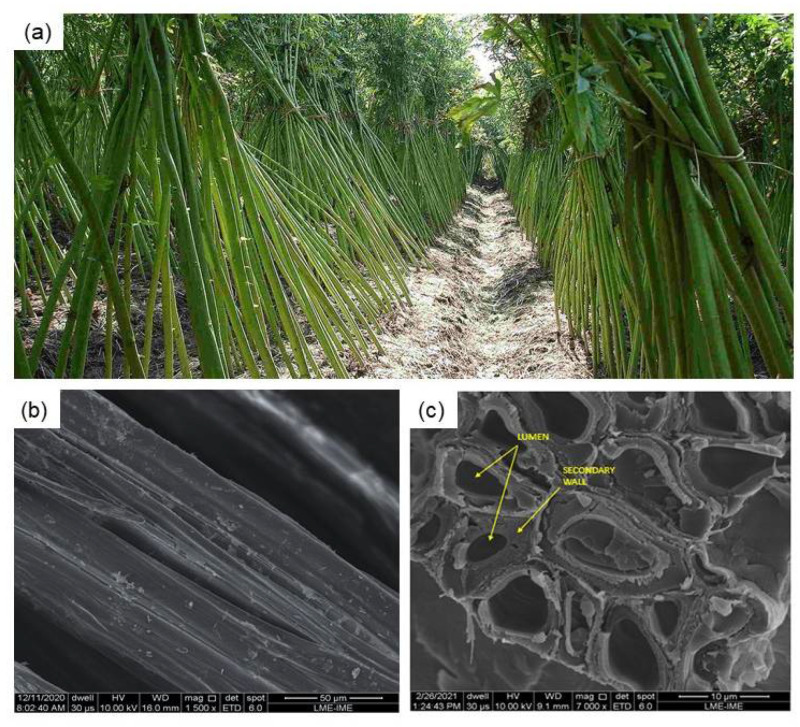
(**a**) Kenaf plants; (**b**) SEM image of a longitudinal section of the kenaf fiber; (**c**) SEM image of a cross-section of the kenaf fiber indicating the lumen and secondary walls of the fiber.

**Figure 2 polymers-13-02016-f002:**
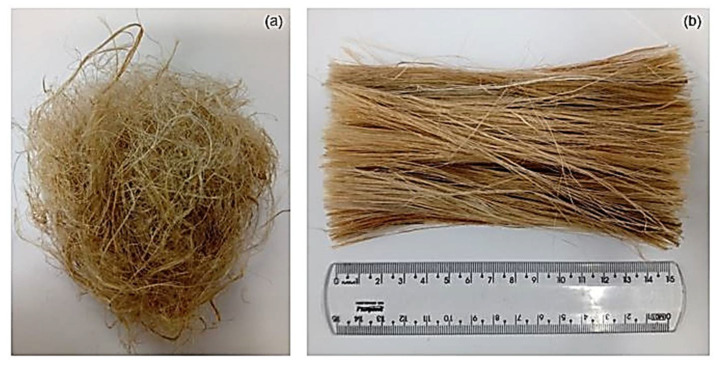
Kenaf fibers (**a**) as received; (**b**) after cleaning and separating the fibers.

**Figure 3 polymers-13-02016-f003:**
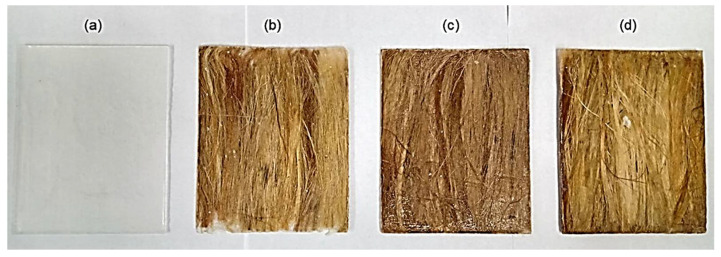
Plates produced: (**a**) 0; (**b**) 10 vol%; (**c**) 20 vol% and (**d**) 30 vol% kenaf fibers.

**Figure 4 polymers-13-02016-f004:**
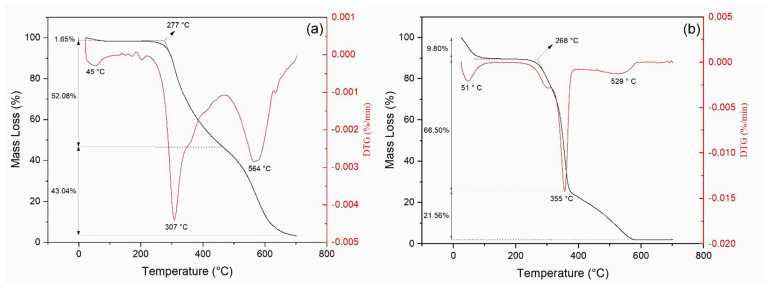
TG and DTG curves for (**a**) epoxy resin; (**b**) kenaf fibers.

**Figure 5 polymers-13-02016-f005:**
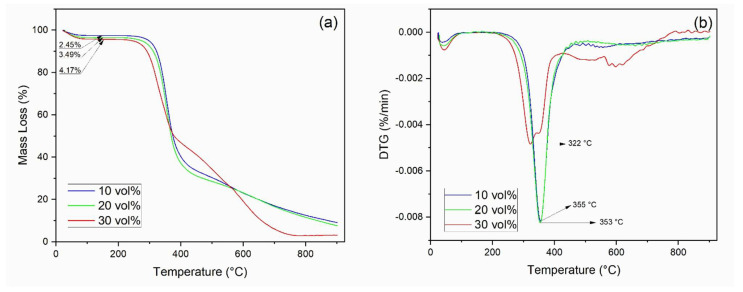
(**a**) TG curves of composites; (**b**) DTG curves of composites.

**Figure 6 polymers-13-02016-f006:**
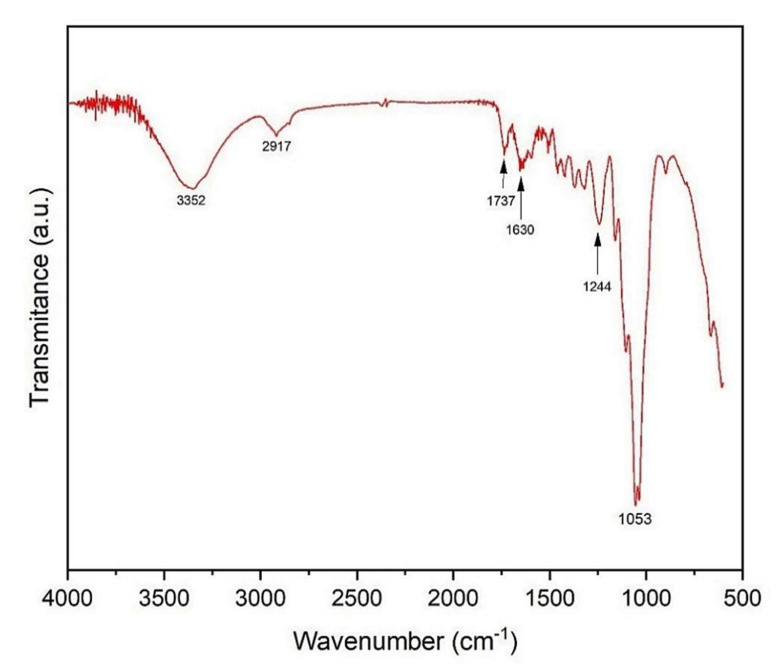
FTIR spectra of kenaf fiber.

**Figure 7 polymers-13-02016-f007:**
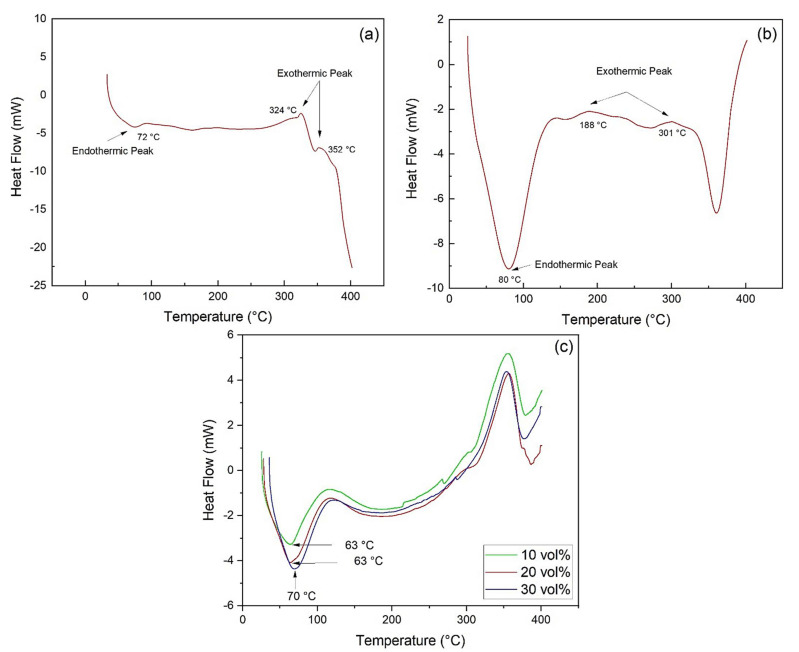
DSC curves: (**a**) epoxy resin; (**b**) kenaf fiber; (**c**) kenaf composites.

**Figure 8 polymers-13-02016-f008:**
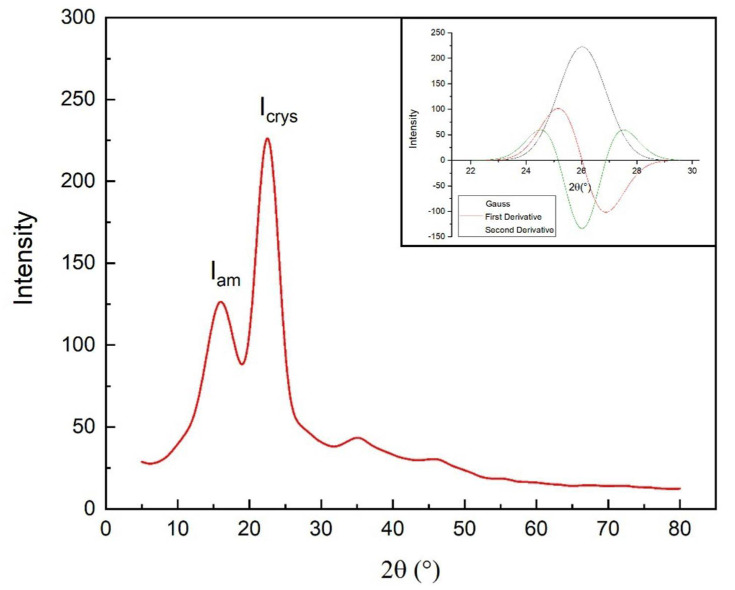
X-ray diffractogram of kenaf fiber. Insertion: deconvolution for calculation of the microfibrillar angle (MFA).

**Figure 9 polymers-13-02016-f009:**
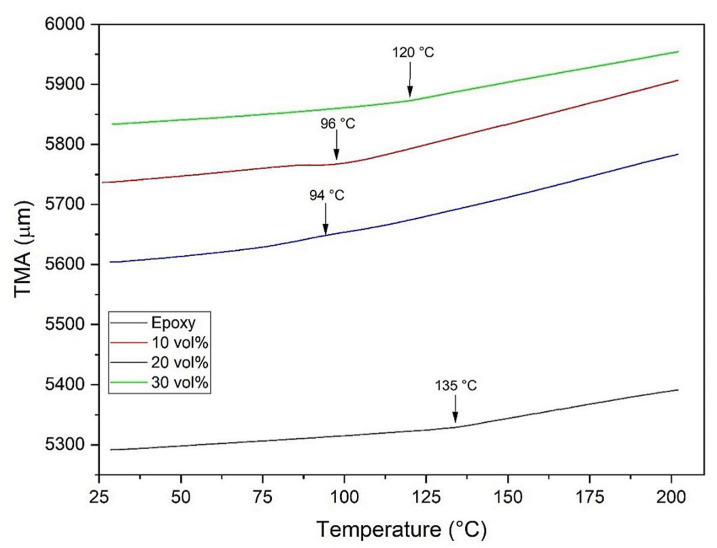
TMA curves obtained for epoxy resin and kenaf composites.

**Table 1 polymers-13-02016-t001:** Properties of the as-supplied kenaf fibers and the DGEBA/TETA epoxy.

Properties	Density (g/cm^3^)	Average Fiber Diameter (µm)	Tensile Strength (MPa)	Young’s Modulus (GPa)	Total Elongation (%)
Kenaf Fiber	1.52 ± 0.28	71.2 ± 6.0	577 ± 71	33.8 ± 3.9	1.7 ± 0.2
Resin Epoxy	1.11 ± 0.05	-	50 ± 8.4	2.7 ± 0.35	2.5 ± 0.4

**Table 2 polymers-13-02016-t002:** Degradation temperatures and mass loss for kenaf fibers and composites.

Sample	T_onset_ (°C)	T_min_ (°C)	T_max_ (°C)	Mass Loss (%)			
				Stage I	Stage II	Stage III	Residue at
Kenaf	216	268	355	9.80	66.50	21.56	2.14 (700 °C)
Epoxy	218	277	307	1.65	52.08	43.92	2.35 (700 °C)
10 vol%	237	302	355	2.45	62.93	25.52	9.10 (900 °C)
20 vol%	235	299	353	3.49	64.41	24.50	7.60 (900 °C)
30 vol%	230	293	322	4.17	51.01	41.63	3.19 (900 °C)

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
