# Peer review of "Thermal and Chemical Characterization of Kenaf Fiber (Hibiscus cannabinus) Reinforced Epoxy Matrix Composites"

_polymers, 2021, doi:10.3390/polym13122016_

Round 1
Reviewer 1 Report
Please present the properties of applied kenaf fibers in the Materials section.
"The fibers were put into the mold in a unidirectional direction" - Figure 4 does ot indicate the unidirectional direction. It is not very regular.
Precisely, how were the composites crushed before TGA analysis?
TGA of fibers was performed before or after pre-processing consisting of washing?
Please refer to the bump above 500 degrees at DTG curve of kenaf fibers.
Figure 6 is missing.
How did Authors determine the onset of thermal degradation? Because it is clear that the decompsition of fibers starts significantly earlier than at 277 degrees.
If char residue is the same for kenaf and epoxy, then please explain its increase for composites. Also please explain why its value is similar for 20 and 30% loading.
If the onset of thermal degradation for composites is in the range of 278-288 degrees then why Authors are writing that "the maximum working temperature for the composite is 200°C"?
Please explain why the increasing amount of kenaf fibers (which are characterized by the lower stability than epoxy) are enhancing thermal stability of composites.
Please explain why if the TGA and DSC analysis are performed under the same atmosphere and with the same heating rates, the difference between temperature positions of peaks attributed to the loss of moisture is so big. For kenaf it is 47 degrees, which is very big.
Similar differences are noted for other peaks, why?
Figure 10 is missing.
" Any factor that can affect the molecular mobility of the material: chemical composition, induction of crystallization by stretching, oxidation, the addition of fillers or a material that has a lower molecular mass, and others, affects the glass transition region of the polymeric material, and on its thermal and mechanical behavior" - and which of these factors are the issue in case of presented composites?
Generally the discussion should be noticeably enhanced.
Author Response
RESPONSE TO REVIEWER
The authors would like to thank the Reviewer for the valuable comments and suggestions on the structure and scientific aspects that contribute to improve the manuscript. Amendments are provided accordingly. Responses to each comment are listed below and all modifications/additions were marked as track Changes in the revised version of the manuscript.
Reviewer #1
Comment (1): Please present the properties of applied kenaf fibers in the Materials section.
Response: as requested, the basic properties of the applied kenaf fibers are now presented in the new Table 1.
Comment (2): "The fibers were put into the mold in a unidirectional direction" - Figure 4 does to indicate the unidirectional direction. It is not very regular.
Response: Although old Figure 4, new Figure 3, does not indicate all kenaf fibers perfectly aligned, base on our experience, this is the best that one might obtain by hand laying up natural fibers in the longest 150 mm direction of the mold. During the introduction of the “fluid” epoxy, some fibers will slightly deviate from their alignment due to the push caused by the flowing liquid. An improved image of Figure 3 with improved quality is now replacing the former Figure 4 to allow a better visualization of the fibers unidirectional alignment.
Comment (3): Precisely, how were the composites crushed before TGA analysis?
Response: The crushing procedure of the composites before TGA analysis is now described in more detail in the revised version.
Comment (4): TGA of fibers was performed before or after pre-processing consisting of washing?
Response: TGA of fibers was performed after the cleaning and drying of the fibers, just in the condition which the kenaf fibers were used to make the composites. This is now clarify in the revised version.
Comment (5): Please refer to the bump above 500 degrees at DTG curve of kenaf fibers.
Response: This is an important DTG event, which was missing in the discussion. The visible bump around 500-600 °C in the kenaf fibers DTG curve is now explained in the revised version.
Comment (6): Figure 6 is missing.
Response: Figure 6 is now clearly shown in the revised version.
Comment (7): How did Authors determine the onset of thermal degradation? Because it is clear that the decomposition of fibers starts significantly earlier than at 277 degrees.
Response: The onset of thermal degradation, after the initial weight loss due to moisture, was determined by standard intercept point of straight lines associated with end of initial stage and beginning of sudden fall of Tg curve. More precise measurements indeed revealed a lower onset of thermal decomposition for the kenaf fiber, as pointed out by the reviewer. This is now indicated in old Figure 5(b), new Figure 4(b), and in the new Table 2.
Comment (8): If char residue is the same for kenaf and epoxy, then please explain its increase for composites. Also please explain why its value is similar for 20 and 30% loading.
Response: These are important questions that are now explained based on the enhanced thermal stability caused by the fiber interaction, which delay the composite total degradation to higher temperatures.
Comment (9): If the onset of thermal degradation for composites is in the range of 278-288 degrees then why Authors are writing that "the maximum working temperature for the composite is 200°C"?
Response: This is an important point raised by reviewer. The reason for our indication of 200 °C as a maximum working temperature is based on required industrial, operational parameter. For safety, this is usually indicated as the closest lower full hundred temperature by considering possible statistical verifications. Now indicated in the revised version.
Comment (10): Please explain why the increasing amount of kenaf fibers (which are characterized by the lower stability than epoxy) are enhancing thermal stability of composites.
Response: Incorporation of natural fiber into polymer matrix does not behave as an additive rule of mixtures for thermal properties. The fiber interaction with macromolecular chains affects the polymer glass transition temperature allowing thermal softening to be displaced higher temperatures. This subject is now discussed in the revised version. Ongoing DMA research work is expected to clear this question.
Comment (11): Please explain why if the TGA and DSC analysis are performed under the same atmosphere and with the same heating rates, the difference between temperature positions of peaks attributed to the loss of moisture is so big. For kenaf it is 47 degrees, which is very big.
Response: Another relevant point raised by the reviewer, as both TGA and DSC are associated with thermal behavior. However, TGA refers to weight loss while DSC depicts heat flow. In fact, in both DTG and DSC curves of kenaf fiber begin to lose weight and dissipate heat right from the beginning of the corresponding test at 20 °C. The maximum DTG weight loss at 51 °C is mostly related to surface humidity. The DTG peak extends up to 100 °C when total moisture is lost. On the other hand, the heat flow from water evaporation is cumulative and reaches a maximum at 80 °C, close to the point where all moisture is vanished. This is explained in the revised version with a new DSC curve for kenaf fiber.
Comment (12): Similar differences are noted for other peaks, why?
Response: As aforementioned, weight loss and heat flow maximum are not necessarily matched in terms of peak temperature. This happens not only for the kenaf fiber, but also for epoxy and composites Explanation is given in the revised version.
Comment (13): Figure 10 is missing.
Response: Figure 10 is now clearly shown in the revised version.
Comment (14): "Any factor that can affect the molecular mobility of the material: chemical composition, induction of crystallization by stretching, oxidation, the addition of fillers or a material that has a lower molecular mass, and others, affects the glass transition region of the polymeric material, and on its thermal and mechanical behavior" - and which of these factors are the issue in case of presented composites?
Response: This is a relevant question that we failed to discuss. In the case of kenaf fibers regarding thermal behaviour the main factor is the addition of fiber as a reinforcing filler to the epoxy matrix restricting the macromolecular mobility. We apologize for omitting this point, which is now discussed in the revised version.
Comment (15): Generally, the discussion should be noticeably enhanced.
Response: As recommended, the discussion of all results is now noticed by enhanced based on the reviewer’s relevant questions.

Reviewer 2 Report
The authors revisited and presented structural, thermal and chemical findings of kenaf fiber/epoxy composites. The manuscript shows sets of different experiment for kenaf fibre/epoxy composites and it may interest reader since there is an increasing attention for bio-based fibres for fibre reinforced composites. However, the submitted manuscript has a poor structure and several flaws. Therefore, the current manuscript cannot be accepted, a major revision is needed. Please satisfy following comments and suggestions:
-Paragraphing is an issue in the whole manuscript please go through the text and apply formal writing standards.
-Line 54 why do authors stress "with a DOI number", if there is no reason please remove and modify the paragraph. Technological relevance?? are Scopus findings an evidence for technological developments? Please remove.
-Personal opinion, figure 2 is not necessary. If will be kept, explain the lines and colour better in the caption.
-Line 62-63 Needs rephrasing.
-Line 64 As a result? As a result of what?
-Line 69-83 Please rephrase the most of the sentences in this paragraph to achieve standards of formal written scientific language. "Xue et al. [58] studied tensile properties." ??? Please be more explanatory.
-Please reference recently published highly related works about adhesion properties kenaf fibre with epoxy resin and environmental assessment of bio-based composite constituents, which will interest potential readers (Materials 2020, 13(9), 2129; https://doi.org/10.3390/ma13092129, Sustainability 2021, 13(3), 1160; https://doi.org/10.3390/su13031160)
-Section 2.1 what is the name of the epoxy resin? - link and reference would be useful for a reader.
-Section 2.2 First issue is there is a density characterisation in comp. processing section. Second issue this section includes results. Structurally this section is methodology and materials, so these problems need to be resolved. Also which standard has been used for density measurement must be given. What are the diameters of the fibres and their SD would be better - please consider giving these properties in a table which may also show it for the resin.
-Please make all the graphs consistent, such as all graphs with inwards ticks or outwards ticks/ uppercase or lowercase inner (a b c)s with same font size . Please also recheck the consistency of plots within same Figure such as axis range in Figure 8.
-Figure 9 Inset graph not clear. Is there particular reason why Y axis does not start from 0 and why 0 of x axis is not given?
-Figure 10 caption and plot do not match. Please also only use English or be consistent with your text.
-If the bullet point conclusion will be kept, please remind the aim and objective of the study and introduce what's been done in the conclusion. It's better for a reader.
-Line 283 how and why superior thermal and mechanical properties? please avoid vague assumption especially in conclusion only provide strong findings of the work.
(please note that the manuscript I had from Editors has different title as "Thermal and Chemical Characterization of Kenaf Fiber (Hibis-2 cus Cannabinus) Reinforced Epoxy Matrix Composites")
Author Response
RESPONSE TO REVIEWER
The authors would like to thank the Reviewer for the valuable comments and suggestions on the structure and scientific aspects that contribute to improve the manuscript. Amendments are provided accordingly. Responses to each comment are listed below and all modifications/additions were marked as track Changes in the revised version of the manuscript.
Reviewer #2
General Comment: The authors revisited and presented structural, thermal and chemical findings of kenaf fiber/epoxy composites. The manuscript shows sets of different experiment for kenaf fibre/epoxy composites and it may interest reader since there is an increasing attention for bio-based fibres for fibre reinforced composites. However, the submitted manuscript has a poor structure and several flaws. Therefore, the current manuscript cannot be accepted, a major revision is needed. Please satisfy following comments and suggestions:
Response: The authors thank the reviewer for the summary and indication of shortcomings. The request major revision was carried out according to the reviewer’s comments and suggestions required for our manuscript acceptance.
Comment (1): Paragraphing is an issue in the whole manuscript please go through the text and apply formal writing standards.
Response: As recommended a through revision is now applied a formal writing standard.
Comment (2): Line 54 why do authors stress "with a DOI number", if there is no reason please remove and modify the paragraph. Technological relevance?? are Scopus findings an evidence for technological developments? Please remove.
Response: The authors agree with the recommended removal and modifications, which have been conducted in the revised version.
Comment (3): Personal opinion, figure 2 is not necessary. If will be kept, explain the lines and colour better in the caption.
Response: By thinking twice, we follow the reviewer opinion and figure 2 is removed in our revised version.
Comment (4): Line 62-63 Needs rephrasing.
Response: Lines 62-63 in the previous version was rephrase for clarity.
Comment (5): Line 64 As a result? As a result of what?
Response: The reviewer is right, the term “As a result” is inadequate and has been changed to “As such” in the revised version.
Comment (6): Line 69-83 Please rephrase the most of the sentences in this paragraph to achieve standards of formal written scientific language. "Xue et al. [58] studied tensile properties.”??? Please be more explanatory.
Response: As recommended, sentences in the indicate paragraph have been rephrased to achieve formal scientific language.
Comment (7): Please reference recently published highly related works about adhesion properties kenaf fibre with epoxy resin and environmental assessment of bio-based composite constituents, which will interest potential readers (Materials 2020, 13(9), 2129; https://doi.org/10.3390/ma13092129, Sustainability 2021, 13(3), 1160; https://doi.org/10.3390/su13031160)
Response: The indicated recently published papers are now cited to support the relevance of our work in terms of sustainability and environmental advantages using a natural fiber like kenaf as biobased composite constituent.
Comment (8): Section 2.1 what is the name of the epoxy resin? - link and reference would be useful for a reader.
Response: As requested, the name of the epoxy resin and corresponding link and reference are now indicated in section 2.1.
Comment (9): Section 2.2 First issue is there is a density characterisation in comp. processing section. Second issue this section includes results. Structurally this section is methodology and materials, so these problems need to be resolved. Also which standard has been used for density measurement must be given. What are the diameters of the fibres and their SD would be better - please consider giving these properties in a table which may also show it for the resin.
Response: The conflict indicated by the reviewer is now taken into consideration by showing the density and fiber diameter measurements in results of a new section 3.1. Also, a new Table presents these kenaf fiber properties as well as the epoxy resin properties.
Comment (10): Please make all the graphs consistent, such as all graphs with inwards ticks or outwards ticks/ uppercase or lowercase inner (a b c)s with same font size . Please also recheck the consistency of plots within same Figure such as axis range in Figure 8.
Response: As requested, all graphs were checked for consistency; particularly in terms of same font size and axis range.
Comment (11): Figure 9 Inset graph not clear. Is there particular reason why Y axis does not start from 0 and why 0 of x axis is not given?
Response: The authors fully agree that insert in old Figure 9, new Figure 8, is not clear and a better graph is now provided clearly indicating the 0 in the axis.
Comment (12): Old Figure 10, new Figure 9, caption and plot do not match. Please also only use English or be consistent with your text.
Response: The authors apologize for inconsistency and mismatch between caption plot in Figure 10, which have been corrected in the revised version.
Comment (13): If the bullet point conclusion will be kept, please remind the aim and objective of the study and introduce what's been done in the conclusion. It's better for a reader.
Response: As suggested, an initial summary of aim and objective of our study with brief introduction of that has been done is now included in the conclusions.
Comment (14): Line 283 how and why superior thermal and mechanical properties? please avoid vague assumption especially in conclusion only provide strong findings of the work.
Response: The authors agree with the reviewer’s comment on vague assumption. More specific modifications are now provided regarding the contribution of our work.
Comment (15): (please note that the manuscript I had from Editors has different title as "Thermal and Chemical Characterization of Kenaf Fiber (Hibis-2 cus Cannabinus) Reinforced Epoxy Matrix Composites")
Response: The different title from the Editors might have a misprinting or misconfiguring. The correct title is “Thermal and Chemical Characterization of Kenaf Fiber (Hibiscus Cannabinus) Reinforced Epoxy Matrix Composites”, which is now shown in the revised version.

Round 2
Reviewer 1 Report
"Comment (7): How did Authors determine the onset of thermal degradation? Because it is clear that the decomposition of fibers starts significantly earlier than at 277 degrees.
Response: The onset of thermal degradation, after the initial weight loss due to moisture, was determined by standard intercept point of straight lines associated with end of initial stage and beginning of sudden fall of Tg curve. More precise measurements indeed revealed a lower onset of thermal decomposition for the kenaf fiber, as pointed out by the reviewer. This is now indicated in old Figure 5(b), new Figure 4(b), and in the new Table 2."
But Authors stated that the fibers were dried prior to the TGA analysis? So how comes they were dried and still show almost 10% of moisture? Also why stability was determined after the initial loss of moisture? If the moisture is part of the fibers then why this step was just omitted?
Also, onset is the beginning of something, just from definition. I am aware that in many works there is this strange term describing the crossing of straight lines, but it is very unprecise. Onset should be determined as a temperature associated with the particular mass loss, depending on the convention it may be 1% or 2%, then it is precise and comparable. Now it can be clearly seen from curves that decomposition starts at lower temperatures than reported in Table.
"The reason for the same amount (12.97%) of residue for both 20 and 30 vol% kenaf fiber composites in Table 2 is still not clear. Repeated TGA tests conducted at higher temperatures (900 °C) are being conducted to clarify this point as part of our ongoing research work" - please present these results here or explain the actual reason for the same amount of residue.
"Comment (8): If char residue is the same for kenaf and epoxy, then please explain its increase for composites. Also please explain why its value is similar for 20 and 30% loading.
Response: These are important questions that are now explained based on the enhanced thermal stability caused by the fiber interaction, which delay the composite total degradation to higher temperatures."
Please describe these interactions and how they may enhance the stability of composite in details.
"Comment (9): If the onset of thermal degradation for composites is in the range of 278-288 degrees then why Authors are writing that "the maximum working temperature for the composite is 200°C"?
Response: This is an important point raised by reviewer. The reason for our indication of 200 °C as a maximum working temperature is based on required industrial, operational parameter. For safety, this is usually indicated as the closest lower full hundred temperature by considering possible statistical verifications. Now indicated in the revised version."
Please discuss in more details this "required industrial, operational parameter". Why it is indicated as the closest lower full hundred temperature? Any reference for that? I believe that in industry the precision is significantly higher than 100 degrees. I see that in the paper it is indicated that it could be increased to 250 degrees, why 250 degrees? Why not 240 or 260 degrees?
Authors cannot write in all issues that they are now cleared in ongoing work, because Authors should discuss clearly results presented in this work, not any further one.
Author Response
The authors would like once again to thank the Reviewer for the valuable comments and suggestions to improve our manuscript. Responses to each new comment are listed below and all modifications marked as track Changes in the new revised version.
Reviewer New Comment [1]: "Comment (7): How did Authors determine the onset of thermal degradation? Because it is clear that the decomposition of fibers starts significantly earlier than at 277 degrees.
Response (7): The onset of thermal degradation, after the initial weight loss due to moisture, was determined by standard intercept point of straight lines associated with end of initial stage and beginning of sudden fall of Tg curve. More precise measurements indeed revealed a lower onset of thermal decomposition for the kenaf fiber, as pointed out by the reviewer. This is now indicated in old Figure 5(b), new Figure 4(b), and in the new Table 2."
But Authors stated that the fibers were dried prior to the TGA analysis? So how comes they were dried and still show almost 10% of moisture? Also why stability was determined after the initial loss of moisture? If the moisture is part of the fibers then why this step was just omitted?
Response [1]: This is a pertinent question, which certainly applies to hydrophobic materials like common polymers. However, the as supplied bundle of kenaf fibers used in the the present work needed to be initially cleaned by immersing in water and then dried to eliminate strange substances and remove impurities. Being extremely hydrophilic, the pure kenaf fibers immediately absorb air humidity, which might be raised to 10% by the time fibers are separated and prepared to be thermogravimetric test or aligned inside the mold to be mixed with the fluid epoxy. This is a common fact reported in most TGA tests with natural fibers (doi:10.1016/j.msea.2012.05.109). To our knowledge, thermal instability does not apply to the loss of surface humidity, which is not an intrinsic part of the fiber, but to the degradation of its structural components including cellulose, hemicellulose, lignin and even the release of chemical hydration water above 100 °C.
Reviewer New Comment [2]: Also, onset is the beginning of something, just from definition. I am aware that in many works there is this strange term describing the crossing of straight lines, but it is very unprecise. Onset should be determined as a temperature associated with the particular mass loss, depending on the convention it may be 1% or 2%, then it is precise and comparable. Now it can be clearly seen from curves that decomposition starts at lower temperatures than reported in Table.
Response [2]: The authors fully agree that onset is the very beginning. However, in a continuously descending TG curve as in our Figure 4(a) for kenaf fiber, it is difficult to precisely identify the transition from the initial low inclination (slope) to the subsequent steeper slope part where structural degradation takes place. That is why the crossing of straight lines (low slope x steeper slope) is used. The Reviewer suggested the onset at conventional 1% or 2% mass loss (analogous to the 0.2% for the yield stress in a tensile curve). This, however, could not be associated with a true onset of thermal degradation but just the percentage of surface humidity loss. In line with the Reviewer concern of a more precise “onset”, we now propose it to be the starting deviation from the initial low “straight” slope (an analogy with the proportional stress limit at the and of elastic regimen in a tensile curve). In Table 2 this is the new “Tonset” while the previous one obtained from the two straight line intercept point is now indicat as “Tmin”, for minimum detected degradation. We hope this would clarify the Reviewer’s relevant question.
Reviewer New Comment [3]: "The reason for the same amount (12.97%) of residue for both 20 and 30 vol% kenaf fiber composites in Table 2 is still not clear. Repeated TGA tests conducted at higher temperatures (900 °C) are being conducted to clarify this point as part of our ongoing research work" - please present these results here or explain the actual reason for the same amount of residue.
Response [3]: The Reviewer is absolutely right. Despite the current difficulties in running experimental work, TGA tests up to 900 °C were performed for the composites, clearing the doubts about final char residues.
Reviewer New Comment [4]: "Comment (8): If char residue is the same for kenaf and epoxy, then please explain its increase for composites. Also please explain why its value is similar for 20 and 30% loading.
Response: These are important questions that are now explained based on the enhanced thermal stability caused by the fiber interaction, which delay the composite total degradation to higher temperatures."
Please describe these interactions and how they may enhance the stability of composite in details.
Response [4]: An additional explanation on the fiber interaction with the matrix macromolecular structure is now provided to explain the enhancement of thermal stability of the composites. As shown in the TMA results, the kenaf fiber incorporation decreases the glass transition temperature and contributes to displace the thermal softening to higher temperatures.
Reviewer New Comment [5]: "Comment (9): If the onset of thermal degradation for composites is in the range of 278-288 degrees then why Authors are writing that "the maximum working temperature for the composite is 200°C"?
Response (9): This is an important point raised by reviewer. The reason for our indication of 200 °C as a maximum working temperature is based on required industrial, operational parameter. For safety, this is usually indicated as the closest lower full hundred temperature by considering possible statistical verifications. Now indicated in the revised version."
Please discuss in more details this "required industrial, operational parameter". Why it is indicated as the closest lower full hundred temperature? Any reference for that? I believe that in industry the precision is significantly higher than 100 degrees. I see that in the paper it is indicated that it could be increased to 250 degrees, why 250 degrees? Why not 240 or 260 degrees?
Response [5]: Based on the now proposed Tonset determined after the response to the Reviewer New Comment [2], the authors believe a maximum working temperature of 200 °C is perfectly coherent with the revised results presented in Table 2. A raise to 220 °C might eventually be considered as an upper limit and satisfy possible industrial requirements.
Reviewer New Comment [6]: Authors cannot write in all issues that they are now cleared in ongoing work, because Authors should discuss clearly results presented in this work, not any further one.
Response [6]: The authors understand and agree with the Reviewer argument. As for TGA tests up to 900 °C to check char residues, new curves are now included for the composites, which fortunately could be anticipated in our “ongoing research work”. As for DMA results, it was not possible to have them ready for the present work. “Ongoing DMA” mention is now deleted in the new revised version of our manuscript.
Reviewer 2 Report
The authors satisfied most of the comments, and the current version of the manuscript can be granted for publication; however, there is still need for English language and paragraphing style check are required.
Author Response
The authors would like once again to thank the Reviewer for the valuable comments and suggestions to improve our manuscript.
Comment: The authors satisfied most of the comments, and the current version of the manuscript can be granted for publication; however, there is still need for English language and paragraphing style check are required.
Response : The authors are thankful for the Reviewer decision and, as required, focused on further English improvements and performed a complete paragraphing check.
Round 3
Reviewer 1 Report
OK after corrections